# Using Assessment Design Decision Framework in understanding the impact of rapid transition to remote education on student assessment in health-related colleges: A qualitative study

Myriam Jaam[1]☯*, Zachariah Nazar[1]☯, Daniel C. Rainkie[1]☯, Diana Alsayed Hassan[2]☯, Farhat Naz Hussain[3]‡, Salah Eldin Kassab[4,5]‡, Abdelali Agouni[3]‡

1 Clinical Pharmacy and Practice Department, College of Pharmacy, QU Health, Qatar University, Doha, Qatar, 2 Department of Public Health, College of Health Sciences, QU Health, Qatar University, Doha, Qatar, 3 Pharmaceutical Sciences Department, College of Pharmacy, QU Health, Qatar University, Doha, Qatar, 4 Faculty of Medicine, Suez Canal University, Ismailia, Egypt, 5 College of Medicine, Gulf Medical University, Ajman, UAE

☯ These authors contributed equally to this work.
‡ These authors also contributed equally to this work
* myriamj@live.co.uk

**Data Availability Statement:** There are ethical restrictions on sharing these data since it contains

## Abstract

Maintaining integrity and validity with online assessment is a significant issue that is well documented. Overt policies encouraging educators to adopt e-Learning and implement digital services coupled with the dramatic change in the education system in response to the challenges posed by COVID-19, has furthered the demand for evidence-based approaches for the planning and delivery of assessments. This study employed the Assessment Design Decision Framework (ADDF), a theoretical model that considers key aspects of assessment design, to retrospectively investigate from a multi-stakeholder perspective the assessments implemented following the rapid transition to remote learning during the COVID-19 pandemic. One-to-one semi-structured interviews were conducted with faculty and students from the Colleges of Pharmacy, Medicine and Health Sciences. After inductive and deductive thematic analysis three major themes were identified. These reflected on the impact of sudden transition on assessment design and assessment plan; changing assessment environment; and faculty-student assessment related interactions which included feedback. The use of a comprehensive validated framework such as ADDF, to plan assessments can improve validity and credibility of assessments. The strengths of this study lie in the innovative adoption of the ADDF to evaluate assessment design decisions from both an educator and student perspective. Further, the data yielded from this study offers novel validation of the use of ADDF in circumstances necessitating rapid transition, and additionally identifies a need for greater emphasis to be attributed to the significance of timeliness of the various activities that are advocated within the framework.

potentially identifying participant information. These restrictions are imposed by Qatar University Institutional Review Board (QU-IRB). Therefore, for access to these data, the corresponding author should be contacted or the QU-IRB office through: QU-IRB@qu.edu.qa or through 0097 44403-5307 with approval reference number: 1283-EA/20.

**Funding:** The author(s) received no specific funding for this work.

**Competing interests:** The authors have declared that no competing interests exist.

## Introduction

Technology-enhanced learning (TEL) is becoming an integral part of the educational process within higher education institutions owing to its benefit in fostering active learning and its flexibility to support student-centred learning. This is reportedly achieved through facilitating a variety of effective and innovative teaching methods such as teaching through collaboration and flipped classrooms [1]. Further, TEL has demonstrated its capacity to enhance the student learning experience both inside the classroom and through distant learning [2,3]. Ensuring assessment integrity in the context of remote education is recognised as a major challenge, for example, verifying that the student completing the assessment has not substituted him/herself for someone else can be significantly difficult. Therefore, many programs offered in an online or blended form, require students to take some key examinations face-to-face. Currently, several technological solutions for remote invigilation software are available that have been implemented with some success, but their use in traditional campus-based programs is yet to be fully investigated [4].

Due to COVID-19 a public health emergency was declared by the World Health Organization requiring preventive measures to be put in place worldwide to curb the spread of the virus. Among these measures was the temporary closure of educational institutions resulting in a sudden, but necessary shift from face-to-face teaching to remote online learning [5–7]. However, there is little information reported on the approach and impact on student assessment following the rapid transition to remote education.

On March 9, 2020, the State of Qatar announced the temporary suspension of classes in all schools and universities as a precautionary measure to limit the spread of COVID-19. As a result, the Health Cluster (College of Pharmacy (CPH), College of Medicine (CMED), College of Health Sciences (CHS) and the newly established College of Dental Medicine (CDM)) at Qatar University took immediate action to switch the mode of teaching to remote online-based learning using a variety of platforms such as Blackboard Collaborate Ultra, WebEx, Microsoft Teams, Echo360 and Zoom. The switch to online-based learning was sudden, occurring in the middle of the semester, thus the majority of courses delivered within the Health Cluster had remaining assessments to be administered. As a result, each college was required to adapt the design of their assessments and re-consider their respective weightings.

The Bearman et al. *Assessment Design Decisions Framework (ADDF)* is a theoretical model that describes six key aspects for consideration when creating or modifying assessments [8]. These include *purposes of assessment*, *context of assessment*, *learner outcomes*, *tasks*, *feedback processes*, and *interactions*. The first three aspects focus on planning and emphasize the necessary considerations when determining the assessment type. *Tasks* and *feedback processes* considers the design of the assessment itself and any feedback associated with it. The final aspect of the framework is associated with the interaction between educators and stakeholders to ensure the assessment design is understood and students are appropriately informed.

Biotshwarelo et al. used this framework prospectively, to guide assessment decision making around the use of online tests [9]. In this study, the framework has been adapted to use retrospectively to investigate the local challenges and processes involved in planning assessments prevalent during the COVID-19 pandemic, and how students and faculty were affected by the change in assessment strategy.

## Materials and methods

### Study design

This study was a qualitative, interpretivist, thematic analysis of in-depth, one-to-one semi-structured interviews.

## Literature search

A search of the literature was conducted to ascertain evidence-based theoretical assessment models on which to frame this study. The results of our literature search yielded many journal articles and texts, which were reviewed for alignment to the research objectives. The Bearman et al. *Guide to the Assessment Design Decisions Framework (ADDF)* was determined by the researchers to be a comprehensive model that addresses the diverse aspects of assessment under investigation in this study [8].

## Setting

This study was conducted between June and August of 2020 in the Health Cluster of Qatar University, the national university in Qatar. The Health Cluster was launched in 2017, to provide close alignment and integration between the health care programs offered by its four colleges: College of Pharmacy (CPH), College of Medicine (CMED), College of Health Sciences (CHS) and the newly established College of Dental Medicine (CDM). At the time of this study, CDM had yet to initiate its programs and therefore was not included.

Interviews were conducted online using Microsoft Teams (MS Teams). MS Teams is a known video conferencing platform which was commonly used among Qatar University students and staff. Due to the COVID-19 pandemic participants and researchers resorted to conducting the interviews from their home setting. To comply with ethical requirements, participants were asked to place themselves in quiet room with minimal distraction within their home. Similarly, researchers used a room with minimal distraction and ensured the use of headphones to maintain confidentiality.

## Data collection

The population of interest were faculty and students within the Health Cluster. Participants were sampled using a representative and purposive method. Participants were identified through the research team and through communication with the respective department heads. Faculty and students from all the three colleges were recruited according to their college affiliation. Moreover, student participants were selected according to their registered course and year of study so to achieve a fair representation of all programs and year groups in the Health Cluster. Inclusion criteria included faculty members teaching within the Health Cluster who were involved with the design and/or delivery of remote teaching sessions with students. Students were eligible if they were enrolled within the colleges of the Health Cluster and have received remote education as part of their university courses. Exclusion criteria included any participant who meets the inclusion criteria but did not provide consent.

Semi-structured interviews were conducted online via Microsoft Teams using audio only at a time chosen by the participants. No compensation was offered for their participation. Their anonymity and confidentiality were ensured.

Two members (ZJ, MJ) of the research team conducted the interviews under private, recorded conditions. The researchers were faculty employed within the Health Cluster with experience in qualitative research methods and in-depth knowledge of the project. Prior to conducting the interviews, the researchers met and discussed the interview guide to ensure consistency. The interview guide was derived from the associated considerations linked to the six categories in the Bearman et al. *Guide to the Assessment Design Decisions Framework* (Fig 1); and further refined by the research team, in light of findings from the literature review and review of the specific research objectives. In addition, the interview guide was sent for evaluation by an expert in the field of qualitative studies and was piloted with one faculty and one

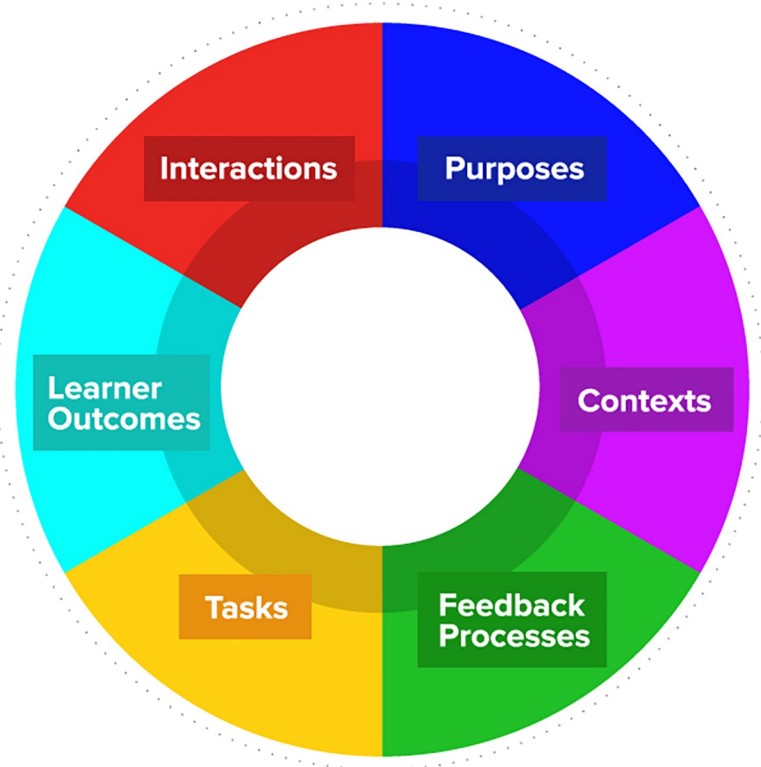

**Fig 1. Assessment Design Decision Framework.**

student. No changes were made to the guide; therefore, the piloted interviews were included as part of the results (S1 File).

Following transcription of the interviews, participants were offered the opportunity to review the transcript; however, in all cases, no modifications were suggested.

## Interview analysis

The audio recordings were transcribed verbatim by the research team (ZJ, MJ, DR, DH, FH); one member (MJ) of the research team reviewed all generated transcripts for accuracy. Data were analysed using an inductive process, then codes and themes were deductively matched based on the *Bearman* framework [8]. The transcripts were divided amongst the four research members (MJ, ZJ, DR, DH) and independently analysed. Transcripts were read through several times to obtain the sense of the whole and then subjected to qualitative thematic analysis following the Braun and Clarke six-step method [10]. Each researcher was allocated a set of transcripts to code. Researchers then met to discuss their coding process. Differences in opinions were discussed until consensus was reached and a coding framework was agreed upon. Codes were considered important according to its match to the ADDF or saliency analysis (i.e. factors that were frequently mentioned, were deemed to be of high relevance by the researchers or participants or had both of these attributes) [11,12]. The four researchers then independently applied the coding framework to a set of three transcripts before meeting to check for consistency in applying the coding framework. Following this, the four researchers then returned to the transcripts they had been allocated and applied the updated coding framework; quotations were identified from the transcripts simultaneously to illustrate each important code [11,12]. In looking for the potential themes, the four researchers met to review the coding

framework, and collated data; this process involved analysing, combining, comparing and mapping how the codes relate to one another, with constant reflection on the six categories articulated in the *Bearman* et al. framework [8]. These themes were then reviewed by the whole research team for their importance in providing significant links between data and addressing key aspects of the research objectives. Throughout this process, a detailed audit trail was kept concerning how the themes were developed and modified. The research team met to name the themes and agree upon a narrative description.

## Ethics

Signed consent forms were sent and collected via email prior to conducting the interviews. Once a participant is identified to meet the inclusion criteria, they were emailed a participant information sheet which included information about the study and a consent form, seven days prior to the interview. Participants were directed to contract the research term if they have any concern or questions during this period. If they agree to participate, they were required to sign the consent form and send it to the principal researcher (MJ). Additionally, before the start of the interview, the participants were reminded of the study objective and confirmation to participate was also obtained. Interview audio recordings were saved for transcribing purposes only after which they were permanently deleted. Consent forms and transcribed data were all kept in Qatar University password protected device in compliance with Qatar University Institutional Review Board (QU-IRB). Institutional ethics approval was obtained from QU-IRB (approval number 1283-EA/20).

## Results

### Participants characteristics

A total of 20 faculty and 31 students were invited to participate in this study, 14 faculty members and 20 students initially agreed to participate. Following correspondence to arrange a time for interview, one faculty member and one student withdrew their consent without providing a reason thus, the study sample consisted of 13 faculty and 19 students. The average interview time was 45 and 47 minutes for faculty and students, respectively. S1 Table summarizes the participants characteristics.

### Results of thematic analysis

Three main themes were identified following thematic analysis of the study data. Each of these themes were further defined into subthemes Fig 2. Table 1 is a representation of the study themes and how they correlate with the ADDF.

   **Theme 1: The impact of rapid transition on assessment design and assessment plan.** This theme predominantly focuses on three main components of *the ADDF* (*purpose*, *learners outcomes*, *and tasks*). The framework describes these components as intention of assessments and its ability to enhance students judgement capability; the integration and alignment of assessments with the learning outcomes; and types of considerations taken in selecting assessments, respectively [8,13]. Accordingly, four subthemes were highlighted within this theme (changes to assessments, learner outcomes, assessment fairness, and administrative support) Fig 2.

   Majority of assessments conducted during this period were designed to generate grades. Nonetheless, there were differences in the changes implemented in each of the colleges. For example, there were no changes made to assessment types in CMED, but assessment weights were modified in some courses. Reports from CPH and Public Health (within CHS), revealed

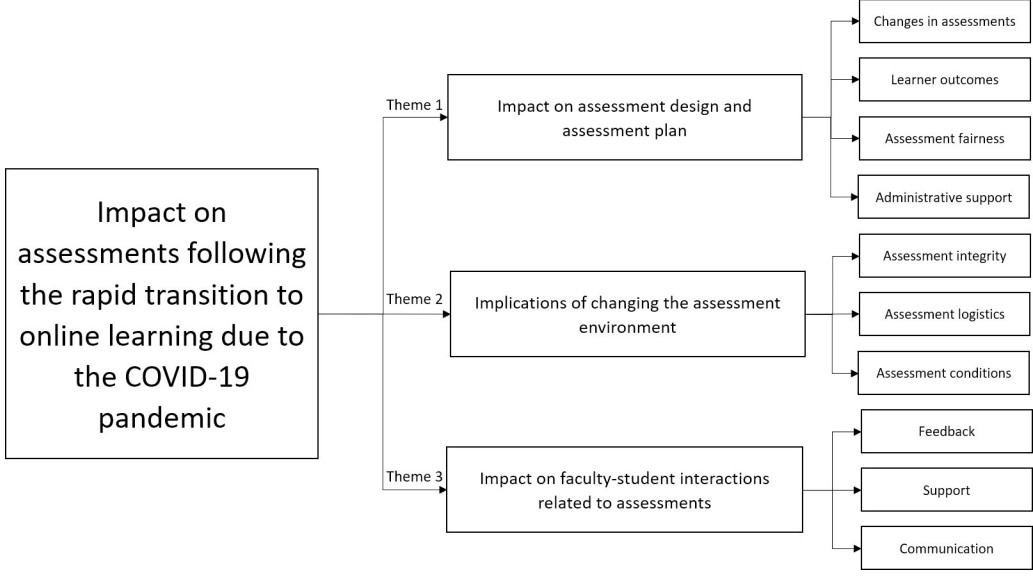

**Fig 2. Themes and subthemes.**

an approach to replace closed-book exams with open-book exams and asynchronous assessments. In line with transitioning to open book exams, multiple choice questions were replaced with open-ended essay-type questions. Final exams were also minimized and sometimes replaced with other forms of assessments such as assignments.

> *'Everything was really heavily weighted, that was the main issue but I think it was necessary because we needed to finish the courses and of course we needed to have equal to 100%' SP4*

Both faculty and students commented on the use of higher Bloom's taxonomy level questions that depend on critical thinking skills in open book exams. Completing such assessments successfully required a good understanding of the content, according to faculty and students. Changes made to assessments created feelings of frustration and uncertainty among some students. Both faculty and students indicated that the number of assessments significantly increased during this period and students felt overloaded with these tasks.

> *'In the beginning of the semester we only had like one assignment for each subject, each course and then after the COVID they, like, they cancelled the exams they added assignments and we have like a lot of pressure' SP11*

**Table 1. ADDF components and corresponding study themes.**

| ADDF component | Brief description | Corresponding theme |
|---|---|---|
| Purpose | Intention of doing the assessment (support student learning, generating grades and equip students to make judgement about their learning) | Theme 1: Impact on assessment design and assessment plan |
| Learners outcome | Integrating the assessment with the learning outcomes including unit outcomes, program outcomes, professional requirements and professional development | |
| Tasks | Types of considerations to be accounted for with assessments such as student's ability to develop and demonstrate their learning | |
| Context | Environmental and personal factors which can impact assessment design | Theme 2: Impact of changing assessment environment |
| Interactions | Involvement of stakeholders in evaluating and improving assessments | Theme 3: Impact on faculty–student interactions |
| Feedback | Assessment related feedback and its influence on designing subsequent assessments | |

In cases where assessment changed to open book, students expressed that they prepared differently for these exams. Additionally, faculty believed that reducing weight of final exams would not be proportional to the efforts expected from students.

*'For the open exam because I know it's an open book, I wasn't serious about the exam like even if I didn't study I can pass.' SP 11*

*'Having a final exam of 10% I don't think it's a proper, let's say assessment to be done to have final worth 10% because I don't think even the student will take it seriously.' FP6*

Along with these changes in assessments in this theme, an evident matter emerged related to faculty workload. Faculty voiced that the effort put into designing new assessment added to their workload.

*'The workload was horrible and any incorporation of new assessments and all that requires time and thinking so we didn't have time for that' FP 2*

It is important to note that administrative support from the organization was provided and guided all decisions related to assessments. Nevertheless, faculty felt that under these guidelines they still had the autonomy to make their own decisions related to their assessments. Faculty felt supported by the university and by their own departments; specifically, new faculty who felt these guidelines were very useful. One faculty also commented on seeking help and support from senior faculty in designing new assessments.

One of the most prominent subthemes was centred around learner outcomes, which is a major component of the *ADDF*. The framework emphasizes that assessments must align with desired learner outcomes. All interviewed faculty indicated that following transition to online assessments, a significant difficulty was assessing student's "soft skills", including communication and presentation skills; and quick clinical-decision making, which can usually be effectively assessed in a classroom environment.

Faculty expressed through the interviews additional multiple challenges they faced in assessing learner outcomes. One of the most common challenges both faculty and students faced was dealing with the dilemma of practical and simulation-based exams. Faculty commented on having to redesign these assessments. For example, the objective structured practical examination (OSPE) that assesses practical lab skills in subjects like anatomy, was conducted online via the electronic platform, Blackboard. Students had to label images instead of physical anatomical structures for the assessment. On the other hand, the objective structured clinical examination (OSCE), which assesses clinical skills, was postponed until next semester for the College of Medicine, while conducted via Zoom for the College of Pharmacy.

*'I was interested to see how after four years they [students] can be presenting their work in front of an audience. Their final work of final project. So unfortunately, we were unable to assess, we were unable to watch this, or, you know, access these skills.' FP 12*

Faculty also discussed the challenge of assessing students fairly and differentiating between students in online assessment. Faculty with previous remote online teaching experience felt more confident in conducting online assessments and felt more comfortable with these changes. Faculty reflected on how assessment questions should be well developed reflecting validity and reliability of the assessment.

*'I didn't care whether the grade would be inflated or not.. I just wanted to have a some sense of justice to those who would have better understanding than those that don't have or those who have studied more than those who didn't' FP 1*

The subtheme assessment fairness revolved around assignment clarity, the provision of rubrics and fair markings, and time allocated to complete assessments. For assignments, many students indicated their need for active support to understand the objectives of the new assignments that were added as part of the new grading scheme. While most students indicated that rubrics were provided for most assessments, students mentioned that in some instances in which rubrics were not used, they were surprised by the grades they received, which, in their opinion, did not accurately reflect their efforts.

*'We were provided with the questions and it was like basically just jump into it you can do whatever you want but for other courses yeah like for example Dr.[name], she wrote a new rubric for exactly what she wants in the grading system and everything and so it was so much easier to follow.' SP 20*

The other factor that both students and faculty commented on was the time allotted to complete assessments. Most students reflected positively on having adequate and fair amount of time; and also on the flexibility and leniency of faculty with assessment time and grading. There were some students however, who indicated that time allocated for the assessment was an issue that affected their assessment completion. One exception was in the College of Medicine, in which faculty did not make major changes related to assessment.

*'The time was good but with many assignments we had this wasn't good, you know, we had to finish the two or three assignments within three days.. this is not logical, we cannot finish this within 24 hours like one assessment that requires me much time like we couldn't do with rush just to submit it before the deadline, so I think the timing for each assignment is enough however, because of many other assessments we couldn't do them well' SP9*

**Theme 2: Implications of changing assessment environment.** The results of this theme were largely centred around contexts of assessment within the *ADDF* [8,13]. The framework describes the *contexts of assessment* to consider including the *characteristics of learners*, *institutional assessment principles and policies*, *professional requirements*, *departmental norms* and *expectations*, the *overall program and purpose of assessment* and *the learning environment*. From the interview results, the authors identified three major subthemes related to the implications of changing the assessment environment including trusting students to complete assessments honestly (integrity); the logistics of assessment deployment and completion; and assessment surroundings and conditions (Fig 2).

Faculty employed a mix of subjective and objective measures to determine if students were completing assessments honestly to validate assessment results. To facilitate these measures, additional assessment logistics and policies were required. While some advice was provided by upper administration (see Theme 1), colleges and course instructors employed different approaches which were dependent on the software and resources available to them.

Many faculty expressed that objective evidence of dishonesty, such as the use of cameras during exams or the use of plagiarism software (e.g. iThenticate, TurnItIn), was fundamentally required to ensure valid assessment results. Students had positive comments about being able to receive feedback about the percentage plagiarised so that they could take the time to improve their paraphrasing but had conflicting opinions about the use of cameras, depending

on the context. Additional methods employed to mitigate cheating for online exams and quizzes included randomizing questions and responses, reducing available exam time, and adopting higher levels of the Bloom's taxonomy. Students believed that cheating during an exam could still occur but were not aware of any such instances.

*'For the online open book exam, yes, so that's a challenge [opening the camera] because, uh, you* cannot *observe the students during the exam. because of the our culture we cannot, uh, you know, uh switch the Cam on'. FP5*

*'I guess making the questions kind of like. . . I don't want to say long. . . but long enough to prevent students from having a lot of extra time at the end to think about sharing answers among each other' FP1*

Regarding assessment logistics, students remarked that the home environment was not conducive to undertaking assessments, due to home distractions that they could not control (e.g. internet failure, family/children in the background). This was contrary to some faculty's beliefs who assumed students should be able to perform in the online environment as well as in the classroom.

*'Sometimes your family might be distracting, someone will knock the door, someone will come inside and they will make noise.. they will not understand that you are taking an exam and you are in a critical situation.' SP11*

All students did note that they were more comfortable at home and enjoyed not having to spend time commuting. Some noted that they did not have the same psychological stress during exams (both positive and negative aspects–not being primed but also not being overwhelmed) which resulted in students finding it difficult to focus. Divergent from faculty assumptions of student technological savviness, students found it difficult to manage several devices at the same time. One student described the need to have one device open to Blackboard lockdown browser (which blocks the opening of all other applications) and another device to connect to an online communication platform (e.g. Blackboard Collaborate, Zoom, Microsoft Teams) to be observed or to ask questions.

*'Yes the online final exams, how to set up the video or the camera. . . to set up your room or workplace, the place you are going to take the exam on, you have to show your hands, your face and all that' SP16*

Students who had used the platform before–during their face-to-face learning–were more comfortable with it than those who did not use it. Faculty noted the need for IT support and teaching assistant support to proctor and deploy online assessments.

*'We usually have our exams online, so the nature of the exams did not differ much. The only thing that was different was the settings and maybe the type of questions.' SP7*

**Theme 3: Rapid transition and faculty-student interactions.** The results of this theme focused on the feedback processes and interactions within the *ADDF* [8,13]. The framework describes feedback processes as opportunities for feedback, who it is provided by, what the feedback consists of and how learner performance could influence later assessments. The interactions section of the framework describes the influence learners have on assessment processes, how learners will understand the requirements of an assessment, and how the results of

these assessments will change future assessments and teaching activities. From the interview results, the authors identified three major subthemes: communication, support, and feedback (Fig 2).

Participants' comments around communication were focused on the communication received at the start of the rapid transition. Students noted an overall negative experience due to a lack of communication and rationale for course changes.

*'They took so long to announce the laws of like the new rules. I feel so that was like the most stressful when they finally said what we were doing. We were OK, but I feel like they spend so much time wasting a lot of time, not saying a single word, and it was their job to say something. So we were not happy then.' SP1*

However, positive statements were identified when students were included in the decision making process such as flexibility in assessment dates to help balance their academic load, meeting with higher administration to be included in assessment decisions, and receiving guidance and further information about the content and methods of the change in assessments. Some faculty remarked that they engaged students where possible but that the final assessment decision making waited until official notice and support was provided by the institution.

*'We were discussing all the assessments with the faculty members. So for example we had Dr. [name] discussing the [course] assessment with us and he was really so helpful. He actually changed the assessment more than once for us so that it can be more suitable for us or easier for us to go through.' SP16*

Once the assessment decisions were in place, students explained that they required support and feedback to adequately prepare for the assessments. Students commented that practice questions, mock exams, or clear descriptions of an assessment (e.g. assessment descriptions, rubrics) had positive impacts on students' perceptions of assessments during a rapid transition.

*'I met with the students online two or three times and we did a lot of revision and we discussed the materials, the difficult concepts, and I answered all the students questions before the exam.' FP5*

Students commented that feedback were important to their learning. Students noted their desire to receive feedback but did not receive it in all instances. Prior to the assessment, students wanted the opportunity to engage with exam content to manage their expectations such as having mock or practice exercises and to receive formative feedback from instructors on their performance. However, faculty reported that the overall workload of teaching and creating or adapting new assessments did not always afford the opportunity to provide formative feedback.

*'Yes, most of the doctors took such point in support into consideration and they gave us a good feedback. I found the individual feedback was the most the most useful for me.' SP9*

From a summative standpoint after assessments, both faculty and students noted that time, assessment load, and policies were major barriers to faculty providing feedback and students engaging with feedback. However, the authors note that the majority of the students provided

examples of both positive and negatives experiences with feedback suggesting heterogenous approaches used by instructors or course coordinators. Students noted that the volume of assessments prevented them from being able to focus on seeking and learning from provided feedback.

> 'For us it [feedback] was just another hurdle, like, you know, when you're going in a race, you just have to keep running and when you pass one hurdle you have to keep running and you don't have to care about whether you pass that hurdle or not' SP8

Students did not engage with feedback if more steps were required to access the feedback (e.g. making appointment with professors or having to read a model answer and analyse their response). When feedback was provided, students preferred direct, individualized feedback.

> 'For other courses they would just post a grade and when you email them they would say like oh schedule like a zoom meeting with me some other time and you can talk about it. But we never have enough time to be honest.' SP20

## Discussion

This is the first reported study to effectively employ the Assessment Design Decision Framework (ADDF) in a retrospective manner to investigate assessment design and delivery. Further, to our knowledge, it is the first study in which the framework has been adopted to successfully explore the experiences and perceptions of multiple stakeholders relevant to the assessment process. Previous studies have demonstrated the value of adopting ADDF to prospectively guide the planning of assessment; however, data and conclusions derived from this study provides useful evidence of how the ADDF can be applied to target specific areas of assessment necessitous of quality improvement as well as identifying focussed opportunities for continued professional development for educators. In employing the ADDF in the context of the rapid transition that was necessary following the COVID-19 pandemic, our findings suggest the need for greater consideration of time-critical elements that are discussed within the six components of the framework. The ADDF is represented as a cyclical structure reflecting the process of assessment planning as a continuum, which is also reflected in the findings of this study, however, little emphasis is currently given to the significance of timeliness of the various activities that are advocated for consideration in this model.

In higher education assessments serve to support students learning through ensuring their attainment of course outcomes; provide a foundation for students to make judgement about their own performance; and to generate grades for purpose of certification and progress [8,13,14]. The purpose of assessments held during the COVID-19 pandemic was to generate an academic standing for students rather than provide formative feedback to inform teaching and learning [15–17]. Even though many universities have opted for eliminating the scale grading system (A-F) and changing their courses to a pass-fail system throughout this period, Qatar University administration favoured maintaining the scale system and has supported their faculty by providing assessment related guidelines. As a consequence of this decision, different approaches needed to be implemented to conduct assessments for generating grades. Nonetheless, faculty interviewed within this project (and across the globe) indicated leniency in grading taking into consideration the pandemic situation and the anxiety caused by the changes in the learning process [18–21]. This leniency can however, contribute to inconsistencies in grading, grading inflation and loss of assessment reliability [22]. Such factors highlight that elements within the ADDF may be compromised due to rapid transition. Consequently,

authors of this project recommend adding "external influencers" within the *Context* component of the ADDF to enhance its applicability and improve assessment reliability.

Most courses eliminated low stakes assessments such as quizzes making the remaining assessments high stakes such as assignments. Changing assessment weighting can impact student performance as indicated by a study conducted by Cohall and Skeete in 2014, which demonstrated that students performance changed when assessments weights were changed from 60% to 40% relative to course work [15]. Although in this study no correlations were made between the students' semester GPA and the identified themes; faculty and students highlighted that there was a slight grade inflation during this period. This inflation can be due to both slight leniency in grading as well as the changes made to assessment weight and type. Moreover, faculty felt it was difficult to assess students' ability to connect topics together due to reduced weight of final exams as well as sometimes replacing them with other assessments. It has been reported that final cumulative assessments have the benefit of encouraging student information retention which is important in medical and health related fields [23]. Therefore, reconsiderations for final exams are warranted if colleges choose to maintain distance online assessments.

As reported in the literature many faculty opted for using open-book examinations or time constrained examinations [24]. Benefits of open book exams is well-established [25,26] including providing students with real-world practice in a controlled environment and typically assessing higher levels of the Blooms taxonomy (apply/analyse/evaluate) [25,26]. However, for students with minimal exposure to this type of assessment, the interviews indicate that they found it more challenging and time consuming to understand and interpret questions. The students who indicated preference for open book examinations claimed that they found it easier to study as they were not memorizing content which for them is very time consuming; while those who preferred closed-book exams found the questions easier to approach and were familiar with its structure. Literature highlights that assessment method significantly influence student's approach to learning [27]. This indicates a need to strengthen students' skills in learning by providing them with enough practice to tackle different assessment designs. Unfortunately, minimal practice was possible during this period but when provided, students appreciated and benefited from these sessions. This supports literature advocating for the use of diverse assessment approaches giving students the opportunity to demonstrate their knowledge and skills across different contexts [28–30].

Farajollahi et al. highlights that the success of online learning in higher education can only be achieved if both faculty and students are ready for the change–from the traditional face-to-face to online [31]. Nonetheless, faculty and students struggled at first to implement and use the online technology for conducting assessments. The use of multiple devices, the disagreements on the use of cameras and the change of platforms all contributed to additional stress to students and faculty. This is a well-recognized challenge as reported in the Institutional Association of Universities Global Surveys which highlighted that having a good infrastructure alone is not sufficient for smooth transition from face to face learning to online learning [32,33], pointing to preparation, tutorials, and practice needed when using new technology. Therefore, significant training and development for both faculty and students is necessary for a smooth transition [34–37]. Additionally, having a contingency plan of action for risk management which can directly be implemented by higher institutions during such difficult times can serve as a cost and time saver as well as reduce stress while maintaining good standards [38]. In a dynamic organization such as that of higher education, having a contingency plan for crisis situations would yield the necessary processes and resources to facilitate preparedness to adapt to such situations can allow for smooth transition while maintaining a full spectrum of activities.

Thereby reducing transition related burden on all stakeholders involved including faculty and students.

The interviews conducted highlighted the importance of maintaining regular communication with students and including them in the planning of assessments. Faculty-student communication was prominent in all three themes within this project. Lack of communication contributed to students worry and concern and some felt excluded from the decision-making process. This is consistent with ADDF where communication is highlighted in all its components predominantly "context" and "interaction" [8]. The literature indicates that students, being the end-users of the educational process, should play a prominent role in informing faculty's decisions to enhance learning activities and not only receive communication related to exams post design [16]. Therefore, their involvement during the rapid change is of no less importance, and should be included in all aspects of assessment planning and delivery.

Feedback is a major component within the *ADDF* [8]. Providing students with formative feedback is essential for their learning as it provides guidance on students' performance [16,39]. Along with rubrics, feedback supports a student-centred approach by allowing students to revise and improve their performance [15]. Despite this, feedback was limited in many instances due to both faculty and students' overload in dealing with the transition to remote education. Considering other modes of delivering meaningful feedback such as providing audio or video feedback and even considering peer feedback can help reduce the burden while maintaining quality education. *Alan Cann* discussed the use of audio feedback in his research targeting undergraduate students who favoured audio feedback over written feedback [40]. He also highlighted that feedback content alone is as important as the timely provision of feedback [40]. Therefore, avoid challenges associated with the learning curve of audio/video feedback provision (which can be timely at the start) faculty development sessions could target training instructors on the use of other modes of feedback delivery keeping in mind that timely individualized feedback can support students in improving their performance [41]. In fact, with online learning this is of more necessity than in traditional learning, since students are not able to meet with the faculty in person, which may contribute to students demotivation [31].

Students' perceived stress during this period and their concerns about grades were not intended to be measured within this project. However, it was highlighted in many interviews from both students and faculty interviews and was evident in all themes. Stress from the current pandemic, the changes made to learning and assessments as well as sitting assessments from home, collectively–in a way or another–affected students' performance. This is consistent with literature findings [42–45]. During this sudden change, there was no mechanism in place to capture students' emotional and psychological stress and consider it in assessment planning. Additionally, high stakes exams added to the level of students' anxiety. This again emphasizes the importance to consider "external influencers" within the *ADDF* as indicated earlier.

## Limitations

The interview data from the different colleges were collated together rather than presenting each college's approach separately. It is important to highlight that there was heterogeneity across the colleges in approaches to assessment before the transition to remote education. This heterogeneity was also reflected on how each college responded to the change. However, this heterogeneity provided a good representation of various assessment approaches utilized by different colleges which enriched the results. Additionally, the project focussed only on one aspect only of the educational process which is assessments; it is recognized that assessments are only part of the learning process and COVID-19 has impacted the entire educational

process and not assessments alone, However, looking at the entire learning process was beyond the scope of this project.

## Conclusions

In addition to the prospective use of Assessment Design Decision Framework, this study demonstrate that the framework can also be used retrospectively to assesses assessment-related experiences not only from faculty, but also from students' perspectives. The findings form this study emphasize the consideration of time-critical components and external influencers within the ADDF especially upon rapid transition from traditional face to face to remote learning. This will maintain assessment validity and reliability. Finally, faculty professional development can target specific areas within the ADDF that were compromised during the transition such as feedback which can be provided through other means–audio/video, therefore, reducing burden on faculty while maintaining quality education.

## Supporting information

**S1 Table. Participants characteristics.**
(DOCX)

**S1 File. Interview guides.**
(DOCX)

## Author Contributions

**Conceptualization:** Myriam Jaam, Zachariah Nazar, Daniel C. Rainkie, Diana Alsayed Hassan, Farhat Naz Hussain, Salah Eldin Kassab, Abdelali Agouni.

**Data curation:** Myriam Jaam, Zachariah Nazar.

**Formal analysis:** Myriam Jaam, Zachariah Nazar, Daniel C. Rainkie, Diana Alsayed Hassan.

**Funding acquisition:** Myriam Jaam.

**Investigation:** Myriam Jaam, Zachariah Nazar, Daniel C. Rainkie.

**Methodology:** Myriam Jaam, Zachariah Nazar, Daniel C. Rainkie, Diana Alsayed Hassan, Farhat Naz Hussain.

**Project administration:** Myriam Jaam, Zachariah Nazar.

**Resources:** Myriam Jaam, Zachariah Nazar, Daniel C. Rainkie, Diana Alsayed Hassan, Farhat Naz Hussain, Salah Eldin Kassab.

**Software:** Myriam Jaam, Zachariah Nazar, Daniel C. Rainkie.

**Supervision:** Myriam Jaam, Zachariah Nazar.

**Validation:** Myriam Jaam, Zachariah Nazar, Daniel C. Rainkie, Diana Alsayed Hassan, Farhat Naz Hussain, Salah Eldin Kassab.

**Visualization:** Myriam Jaam, Zachariah Nazar, Daniel C. Rainkie, Diana Alsayed Hassan, Salah Eldin Kassab, Abdelali Agouni.

**Writing – original draft:** Myriam Jaam, Zachariah Nazar, Daniel C. Rainkie, Diana Alsayed Hassan, Farhat Naz Hussain, Salah Eldin Kassab, Abdelali Agouni.

**Writing – review & editing:** Myriam Jaam, Zachariah Nazar, Daniel C. Rainkie, Diana Alsayed Hassan, Farhat Naz Hussain, Salah Eldin Kassab, Abdelali Agouni.

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
