## [Decision Letter · Decision Letter 0]

6 May 2021

PONE-D-21-07338

Using Assessment Design Decision Framework in understanding the impact of Rapid Transition to Online-learning on Student Assessment in Health-related Colleges: A Qualitative Study

PLOS ONE

Dear Dr. Jaam,

Thank you for submitting your manuscript to PLOS ONE. After careful consideration, we feel that it has merit but does not fully meet PLOS ONE’s publication criteria as it currently stands. Therefore, we invite you to submit a revised version of the manuscript that addresses the points raised during the review process.

We look forward to receiving your revised manuscript.

Kind regards,

Tareq Mukattash

Academic Editor

PLOS ONE

Journal Requirements:

2a) If there are ethical or legal restrictions on sharing a de-identified data set, please explain them in detail (e.g., data contain potentially identifying or sensitive patient information) and who has imposed them (e.g., an ethics committee). Please also provide contact information for a data access committee, ethics committee, or other institutional body to which data requests may be sent.

2b) If there are no restrictions, please upload the minimal anonymized data set necessary to replicate your study findings as either Supporting Information files or to a stable, public repository and provide us with the relevant URLs, DOIs, or accession numbers. Please see http://www.bmj.com/content/340/bmj.c181.long for guidelines on how to de-identify and prepare clinical data for publication. For a list of acceptable repositories, please see http://journals.plos.org/plosone/s/data-availability#loc-recommended-repositories.

Reviewers' comments:

Reviewer's Responses to Questions

**Comments to the Author**

1. Is the manuscript technically sound, and do the data support the conclusions?

Reviewer #1: Yes

2. Has the statistical analysis been performed appropriately and rigorously? 

Reviewer #1: Yes

3. Have the authors made all data underlying the findings in their manuscript fully available?

Reviewer #1: Yes

4. Is the manuscript presented in an intelligible fashion and written in standard English?

Reviewer #1: Yes

5. Review Comments to the Author

Reviewer #1: Thank you for the opportunity to review this paper

This paper just need few revisions

Please, change online- learning to remote education

Could you give more information regarding the setting where the data was collected

More information regarding informed consent and how you maintain the research ethics is required

Could you explain more regarding recruitment and data collection

More information is required regarding inclusion criteria for the participants

Take a look for a study in the area such as

Abuhammad S. Barriers to distance learning during the COVID-19 outbreak: A qualitative review from parents’ perspective. Heliyon. 2020 Nov 10:e05482.

6. PLOS authors have the option to publish the peer review history of their article (what does this mean?). If published, this will include your full peer review and any attached files.

Reviewer #1: No

---

## [Author Response · Author response to Decision Letter 0]

9 May 2021

Reviewer: This paper just need few revisions

Please, change online- learning to remote education

Response: The change was made in the title and throughout the paper when applicable. 

Reviewer: Could you give more information regarding the setting where the data was collected

More information regarding informed consent and how you maintain the research ethics is required

Response: More information was added about the setting within the methodology. The setting reflected that of the online environment where the interviews took place. 

The following section was added “Interviews were conducted online using Microsoft Teams (MS Teams). MS Teams is a known video conferencing platform which was commonly used among Qatar University students and staff. Due to the COVID-19 pandemic participants and researchers resorted to conducting the interviews from their home setting. To comply with ethical requirements, participants were asked to place themselves in quiet room with minimal distraction within their home. Similarly, researchers used a room with minimal distraction and ensured the use of headphones to maintain confidentiality.” 

Additionally, more information was added under the Ethics section. “Once a participant is identified to meet the inclusion criteria, they were emailed a participant information sheet which included information about the study and a consent form, seven days prior to the interview. Participants were directed to contract the research term if they have any concern or questions during this period. If they agree to participate, they were required to sign the consent form and send it to the principal researcher (MJ). Additionally, before the start of the interview, the participants were reminded of the study objective and confirmation to participate was also obtained. Interview audio recordings were saved for transcribing purposes only after which they were permanently deleted. Consent forms and transcribed data were all kept in Qatar University password protected device in compliance with Qatar University Institutional Review Board (QU-IRB).”

Reviewer: Could you explain more regarding recruitment and data collection

More information is required regarding inclusion criteria for the participants

Take a look for a study in the area such as

Abuhammad S. Barriers to distance learning during the COVID-19 outbreak: A qualitative review from parents’ perspective. Heliyon. 2020 Nov 10:e05482

Response: More information was added into the manuscript “Participants were identified through the research team and through communication with the respective department heads.” and “Inclusion criteria included faculty members teaching within the Health Cluster who were involved with the design and/or delivery of remote teaching sessions with students. Students were eligible if they were enrolled within the colleges of the Health Cluster and have received remote education as part of their university courses. Exclusion criteria included any participant who meets the inclusion criteria but did not provide consent.”

The study suggested by the reviewer reflects on the parents perceptions and barriers to remote education. The reference was reviewed and found to add benefit within the discussion.

---

## [Editor Report · Decision Letter 1]

28 Jun 2021

Using Assessment Design Decision Framework in understanding the impact of rapid transition to remote education on student assessment in health-related colleges: A qualitative study

PONE-D-21-07338R1

Dear Dr. Jaam,

We’re pleased to inform you that your manuscript has been judged scientifically suitable for publication and will be formally accepted for publication once it meets all outstanding technical requirements.

Kind regards,

Tareq Mukattash

Academic Editor

PLOS ONE
---

## [Editor Report · Acceptance letter]

30 Jun 2021

PONE-D-21-07338R1 

Using Assessment Design Decision Framework in understanding the impact of rapid transition to remote education on student assessment in health-related colleges: A qualitative study 

Dear Dr. Jaam:

I'm pleased to inform you that your manuscript has been deemed suitable for publication in PLOS ONE. Congratulations! Your manuscript is now with our production department. 

Kind regards, 

on behalf of

Dr. Tareq Mukattash 

Academic Editor

PLOS ONE